# Genetic landscape of preterm birth due to cervical insufficiency: Comprehensive gene analysis and patient next-generation sequencing data interpretation

**Ludmila Volozonoka**[1]*, **Dmitrijs Rots**[1], **Inga Kempa**[1], **Anna Kornete**[2], **Dace Rezeberga**[2], **Linda Gailite**[1‡], **Anna Miskova**[2‡]

**1** Scientific Laboratory of Molecular Genetics, Riga Stradins University, Riga, Latvia, **2** Department of Obstetrics and Gynecology, Riga Stradins University, Riga, Latvia

‡ These authors share first authorship on this work.

* ludmilavolozonoka@gmail.com

**Data Availability Statement:** All relevant data are within the paper and its Supporting Information files.

## Abstract

Preterm delivery is both a traumatizing experience for the patient and a burden on the healthcare system. A condition distinguishable by its phenotype in prematurity is cervical insufficiency, where certain cases exhibit a strong genetic component. Despite genomic advancements, little is known about the genetics of human cervix remodeling during pregnancy. Using selected gene approaches, a few studies have demonstrated an association of common gene variants with cervical insufficiency. However, until now, no study has employed comprehensive methods to investigate this important subject matter. In this study, we asked: i) are there genes reliably linked to cervical insufficiency and, if so, what are their roles? and ii) what is the proportion of cases of non-syndromic cervical insufficiency attributable to these genetic variations? We performed next-generation sequencing on 21 patients with a clinical presentation of cervical insufficiency. To assist the sequencing data interpretation, we retrieved all known genes implicated in cervical functioning through a systematic literature analysis and additional gene searches. These genes were then classified according to their relation to the questions being posed by the study. Patients' sequence variants were filtered for pathogenicity and assigned a likelihood of being contributive to phenotype development. Gene extraction and analysis revealed 12 genes primarily linked to cervical insufficiency, the majority of which are known to cause collagenopathies. Ten patients carried disruptive variants potentially contributive to the development of non-syndromic cervical insufficiency. Pathway enrichment analysis of variant genes from our cohort revealed an increased variation burden in genes playing roles in tissue mechanical and biomechanical properties, i.e. collagen biosynthesis and cell-extracellular matrix communications. Consequently, the proposed idea of cervical insufficiency being a subtle form of collagenopathy, now strengthened by our genetic findings, might open up new opportunities for improved patient evaluation and management.

**Funding:** The study was supported by a Riga Stradins University Internal Study Grant (to LV). The funders had no role in study design, data collection and analysis, decision to publish, or preparation of the manuscript.

**Competing interests:** The authors have declared that no competing interests exist.

# Introduction

In order to carry a successful term pregnancy, different organs such as the uterus, cervix, placenta, and amniotic membranes as well as the fetus itself must cohesively interact and create a healthy symbiotic relationship with each other and the rest of the female body [1]. However, preterm birth (PTB) remains the leading cause of perinatal morbidity, mortality, and hospitalization in the first year of life in the developed world. Approximately 5–12% of newborns worldwide are born preterm (<37 weeks of gestation) [2]. Prematurity is a tremendous burden on the healthcare system as outcomes are associated with disability-specific lifetime medical, special education, and lost productivity costs [3].

A common phenotype of spontaneous PTB is primarily characterized by progressive cervical effacement, after which preterm premature rupture of membranes (PPROM), persistent uterine contractions, prolapsed fetal membranes, or uterine bleeding may be the reason for acute care seeking.

## Isolated cervical insufficiency

A distinguishable medical condition in obstetrics in which the cervix spontaneously starts to dilate (open) and efface (become thinner) in the absence of the signs and symptoms of labor is cervical insufficiency. The cervix, a collagen-rich organ, must remain closed during pregnancy yet simultaneously undergo a progressive physiological remodeling to prepare for the birth. Physiological cervical remodeling along with uterine contractile activation are the two key events facilitating the birth of a child [4]. This remodeling can be loosely divided into four overlapping phases: 1) softening beginning in early pregnancy, 2) ripening shortly before the birth, 3) dilation starting with the onset of regular uterine contractions and resulting in cervical opening to allow passage of the term fetus, and 4) postpartum repair [4–6]. In cases of cervical insufficiency, dilation of the cervix occurs without painful uterine contractions, leading to inability of the cervix to retain a term pregnancy. Repeating in consecutive pregnancies, cervical insufficiency is one of the causes of recurrent pregnancy loss [7] and can be a serious obstacle to the birth of a healthy child and complication-free postpartum period for the mother and newborn. In contrast, failure of the cervix to dilate would result in unsuccessful parturition [8].

Clinically relevant isolated cervical insufficiency occurs in about 1–2% of all pregnancies, but is associated with as much as 5–15% of pregnancy losses in the second trimester [9,10]. However, as one of the factors in a complex PTB context, the condition is found much more frequently. In 2011, routine recording of cervical ripening was recommended by the Global Alliance to Prevent Prematurity and Stillbirth [11], since a short cervix (defined as a transvaginal sonographic cervical length ≤25 mm in the mid-trimester of pregnancy) is the best predictive factor for spontaneous PTB <34 weeks of gestation in both singletons and twins [12]. The shorter the cervix, the higher the risk; cervical insufficiency is likely at the extreme of this continuum [13].

Multiple factors such as age, inflammation, stress, nutrition, physical activity, socio-economic status, vaginal microbiome, and uterine anomalies affect PTB [14–16]. Mid-trimester cervical weakness may be associated with a variety of events, e.g. cervical ablation (cryo, laser, or electro) or excision (knife, laser, or loop-electrosurgical), cervical intraepithelial neoplasia *per se*, cervical hypoplasia after diethylstilbestrol, or intrauterine infections [17].

PTB is currently perceived as a frequent complex medical condition which corresponds to the concept of a multifactorial disorder [18]–analogous to, for example, cardiovascular disease–the development of which depends on a number of interacting factors including environmental and genetic. Familial aggregation is evident in prematurity [19,20], including cases of

cervical insufficiency, with up to 27% of patients having a first-degree relative with the same diagnosis on the mother's side [21,22]. By contrast, the risk appears to be unaffected by a history of prematurity in the partner's family [23]. Epidemiological data show that fetuses/neonates with Ehlers-Danlos syndrome (EDS), osteogenesis imperfecta, and restrictive dermopathy are at an increased risk of adverse pregnancy outcomes including PTB, PPROM, and cervical insufficiency [24,25]. A few studies have demonstrated a positive association of common gene variants in the mother's genome with cervical insufficiency [22,26,27].

## Current understanding of genetics of cervical remodeling during pregnancy is limited

Prior to the era of '-omics', the majority of studies investigating the role of genetics in prematurity targeted candidate genes with known biological roles potentially related to processes occurring during pregnancy [18,28]. For example, common allelic variants/polymorphisms in *TNF*, *IL1B* and *IL6* genes have most consistently been associated with PTB [29], underlining the role of inflammation in the pathogenesis of prematurity.

More recent studies on the genetics of PTB in humans can be roughly divided into two major categories. The first group is comprised of a small number of large-scale genomic studies investigating possible genetic risk factors for preterm delivery [30–33]. Unfortunately, none of these studies has addressed PTB as a result of cervical insufficiency. The second group consists of transcriptomic studies evaluating differential gene expression during different stages of gestation/parturition in eventless gestations [34–36] and ones of particular phenotypes, e.g. cases of PTB (HP:0001622) or PPROM (HP:0001788; OMIM:610504) [37,38]. For a comprehensive evaluation, please refer to the excellent systematic review and meta-analysis of [39].

Although the largest number of studies has focused on idiopathic PTB, this phenotype should be considered with caution since preterm delivery often encompasses cervical insufficiency, PPROM, placental abruption, uterine overdistension, or a combination of these complications [13]. This idea is further supported by a meta-analysis of gene expression studies across distinct gestational tissues and clinical phenotypes which demonstrated a limited overlap of genes identified as differentially expressed across the studies [39]. This suggests possible different physiological mechanisms underlying each phenotype and also indicates that large gaps still exist in the design of transcriptomic studies in prematurity.

Furthermore, in order to attain the true transcriptomic signature of a certain phenotype, the tissue of study should be chosen wisely. Considering the known heterogeneity of certain tissues, even the biopsy site may have an impact on the results. For example, a recent study has highlighted that it remains to be resolved whether a PPROM signature can be determined in the cervix as the gene expression patterns in cervical biopsies of PPROM in comparison to preterm labor samples did not share cluster membership, suggesting a distinct genetic signature specific to PPROM pathology [37]. Nonetheless, the authors proposed the notion that the rupture of membranes might be accelerated through PPROM-specific remodeling events within the cervix [37]. Of note, cervical insufficiency often results in unscheduled PPROM as well. Moreover, similar to the cervix, the mechanical strength of the fetal membranes is mainly ensured by the collagen network [40,41]. Indeed, collagen types I, III, IV, V, and VI, to name but a few, have been localized in both cervical (derived from the two paramesonephric ducts during embryogenesis) and fetal membrane (derived from the outer trophoblast layer of the implanting blastocyst) tissue [42,43].

Without doubt, our current understanding of human cervix remodeling in pregnancy is limited [1]. This may be the reason for the bias of studied genes in relation to cervical

insufficiency and the surprisingly little information that presently exists on the genetics of pathological cervical remodeling during pregnancy.

Since common variants detectable by genome-wide association studies (GWAS) typically explain only a minor proportion of the heritability of complex diseases [44], there is a hypothesis that the rare variants in multiple genes implicated in PTB may cumulatively contribute to the predisposition of delivering preterm [15,45]. We decided to test this hypothesis by performing next-generation sequencing (NGS) of the DNA of females with a positive anamnesis of isolated non-syndromic cervical insufficiency.

Due to the lack of knowledge of genes implicated in cervix functioning, we also conducted a systematic literature analysis to derive all possible studies on the genetics of the cervix. We subsequently composed a list of genes that play a role in the normal and pathological biology of the cervix in relation to pregnancy and prematurity and used this obtained knowledge to assist our NGS data interpretation. Given the described heritability of cervical insufficiency, the main questions we addressed in this study were: i) are there genes reliably linked to cervical insufficiency and, if so, what are their roles? and ii) how many cases of isolated non-syndromic cervical insufficiency are attributable to these genetic variations?

## Materials and methods

### Identification of genes playing a role in the biology of the cervix

**Systematic literature analysis: Search strategy and study selection.** We conducted a literature search according to the PRISMA guidelines [46] (Fig 1). The screening strategy aimed to retrieve studies focusing on genetic research of defective uterine cervix functioning leading to cervical insufficiency, preterm delivery, or pregnancy loss, as well as records on functional studies addressing the differential expression of genes within the cervix during different stages of normal/compromised pregnancy/parturition.

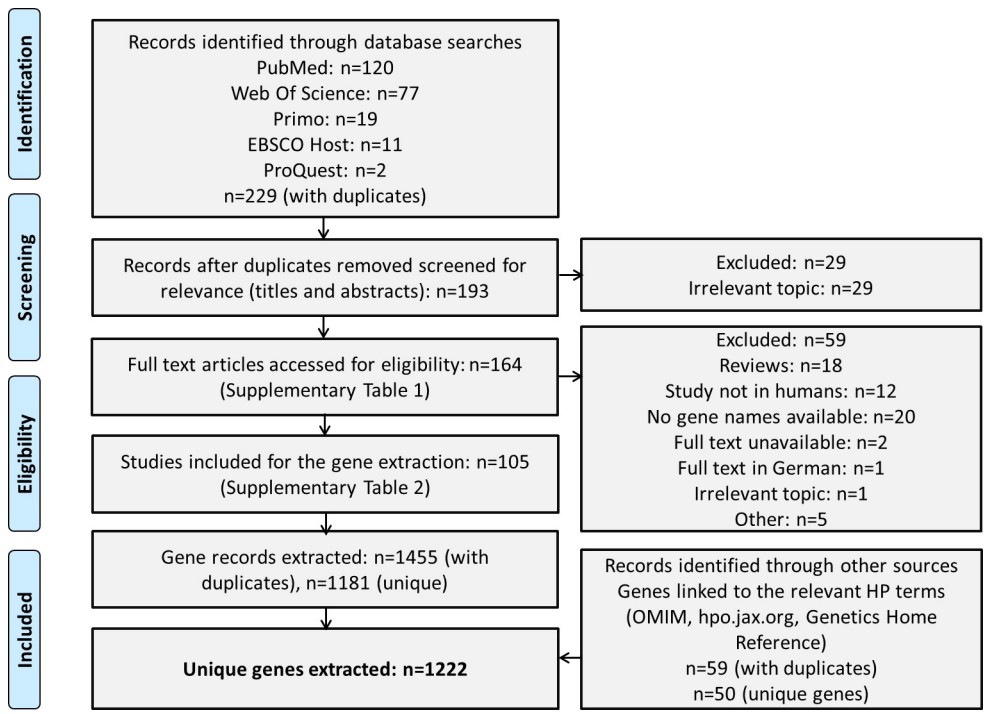

**Fig 1. PRISMA flowchart depicting the literature search and gene extraction.**

The search was performed using the MeSH term "Uterine Cervical Incompetence" (D002581, which is indexed under two higher categories in the MeSH hierarchy: "Pregnancy Complications" and "Female Urogenital Diseases") OR the following keywords: "precocious cervical ripening", "cervical weakness", "cervical insufficiency", "istmocervical insufficiency", "cervical incompetence", "uterine cervix" AND "gene", "genetics", "gene expression", "gene transcription", "transcriptome" AND Humans [Mesh] NOT "cancer", "microbiome", "papilloma". The search was performed in PubMed, EBSCO Host database, Web of Science, ProQuest database, and Primo database. The search was performed in parallel by two reviewers on December 16, 2019, without further restrictions on the publication date.

Inclusion criteria: Study published in a peer-reviewed journal; Study presents original data; Study concentrates on finding a genetic cause of cervical insufficiency and/or preterm delivery; Study concentrates on functional gene analysis of physiological cervical ripening, cervical insufficiency, and/or preterm delivery as a source using cervical tissues. Only human studies were included.

Exclusion criteria: Study concentrates on miscarriage and/or the first trimester of pregnancy; Study concentrates on microRNA, lncRNA, cell-free DNA, ribosomal DNA, cervico-vaginal microbiome, cancer analysis; Study is not in humans; Study is not available in English (S1 Table).

From the eligible papers presenting original data (S2 Table), we extracted the following: gene symbols in a HUGO Gene Nomenclature Committee-approved manner; recorded patient phenotypes, i.e. condition, or relevancy to our study's questions if healthy individuals were analyzed; biological material used for the analysis (DNA, cervical biopsy, other); type of study, e.g. functional, association, or other; information on the gene selection approach (unbiased genome-wide or selected gene approach). Screening of all the reviews (n = 18) for original references did not yield any additional articles to those in the original search list.

**Additional gene identification.** Further, we searched the most relevant Human Phenotype Ontology (HPO) terms for association with certain genes as well as all EDS-, osteogenesis imperfecta-, and restrictive dermopathy-related genes through Online Mendelian Inheritance in Man (OMIM), Genetics Home Reference (GHR), and https://hpo.jax.org as they previously showed clinical association with cervical insufficiency. The following HP terms were screened: Premature birth following premature rupture of fetal membranes (HP:0005100); Premature rupture of membranes (HP:0001788); Premature birth (HP:0001622); Premature delivery because of cervical insufficiency or membrane fragility (HP:0005267); Uterine rupture (HP:0100718); Uterine prolapse (HP:0000139); and Cervical insufficiency (HP:0030009).

## Gene analysis

Based on the data obtained from all the eligible studies and additional syndromic gene searches, we composed three different lists of genes according to their relation to the genetics of the cervix. The first list encompassed genes primarily linked to cervical insufficiency and were either: i) studied directly in relation to cervical insufficiency alone or with any other obstetrical condition; ii) shown to have an association with cervical insufficiency in cases of any genetic syndrome; or iii) shown to have an established gene-phenotype relationship with cervical insufficiency (HP:0030009; HP:0005267) as identified through the HPO/OMIM searches.

The second list contained genes with less evidence for cervical insufficiency than those in the first list and/or alleged collagen-related associations and were either: i) studied in relation to PPROM alone or with any other obstetrical condition; ii) shown to have an established gene-phenotype relationship with obstetrical complications (HP:0005100; HP:0001788;

HP:0100718; HP:0000139; HP:0000140; HP:0001622 –not alone) as identified through the HPO/OMIM searches; iii) known to cause a genetic syndrome clinically associated with cervical insufficiency; or iv) studied in relation to cervical insufficiency but having no association as shown from case-control studies.

The third list consisted of genes demonstrating a function within the uterine cervix as shown from functional gene studies of physiological cervical ripening/pregnancy/parturition using either cervical biopsies/swabs from females without obstetrical complications or tissue cultures.

Additionally, to identify any differences in the biological information of genes studied exploiting selected gene approaches in comparison to those studied using unbiased approaches (i.e. genome-wide studies), we functionally annotated genes from both groups using the ConsensusPathDB interaction database [47] gene set analysis function 'over-representation analysis' and looked for 'Pathway-based sets' in all built-in pathway databases and 'Gene ontology categories' (level 2 categories) with a p-value cut-off of 0.01.

## Next-generation sequencing of patients with cervical insufficiency

**Ethical considerations.**   The study was conducted in accordance with the Declaration of Helsinki's ethical principles. The study protocol was approved by the governmental Central Medical Ethics Committee (Nr.2/18-03-21). Patients considered for genetic testing were counseled and the testing principles were explained. All the patients recruited signed an informed consent.

**Subjects.**   The study recruited 21 females of Caucasian ethnicity (attending Riga Maternity Hospital between 2017 and 2019) with presentation of painless cervical dilatation in the ongoing pregnancy (as identified during a standard cervical length measurement using transvaginal ultrasound between the 18th and 22nd week of gestation) and/or a positive anamnesis of pregnancy loss and/or preterm delivery due to cervical insufficiency without contractions in singleton pregnancies, and the absence of diagnosed genetic conditions. Vaginal infection (exclusion criterion) was ruled out by a pH assessment, where pH>4.4 indicated the presence of infection.

Among the recruited females, the total number of pregnancies excluding legal abortion, indicated medical abortion, and extra-uterine pregnancies was 3.5±2.2 (TP-OP group, Table 1). Out of those, 52% resulted in late pregnancy loss (LPL, >12 weeks <22 weeks) or PTB (<37 weeks)–a group most likely associated with cervical insufficiency. Early pregnancy losses (EPL, <12 weeks) were separated as they are mostly related to fetal chromosomal abnormalities. Six out of the 21 patients experienced PPROM in one of their pregnancies. Table 1 was completed after the outcome of each patient's ongoing pregnancy was resolved.

**Next-generation sequencing analysis.**   Genomic DNA was isolated from whole peripheral blood using an adapted phenol-chloroform extraction. NGS analysis was carried out using Illumina's TruSight One Sequencing Panel Capture Kit (USA) covering all genes currently reviewed in the clinical research setting (4810 genes) and generating indexed paired end

**Table 1. Baseline demographic and clinical characteristics of participants.**

| Age, years | Weight, kg | Height, m | BMI, kg/m$^2$ | TP | OP | TP-OP | EPL | LPL+PTB | CL, cm |
|---|---|---|---|---|---|---|---|---|---|
| 35±4.8 | 73.2±16.7 | 1.7±0.05 | 26±5.5 | 4.5±2.5 | 1.0±1.1 | 3.5±2.2 | 0.5±1.0 | 1.9±1.7 | 1.53±0.5 |

BMI–Body Mass Index; TP–Total Pregnancies; OP–Other Pregnancies including legal abortion, indicated medical abortion and extra-uterine pregnancies; TP-OP–Total Pregnancies excluding OP; EPL–Early Pregnancy Loss (<12 weeks); LPL+PTB–Late Pregnancy Loss (>12 weeks <22 weeks) and Preterm Birth (<37 weeks); CL–Cervical Length.

(2×75) reads. Template DNA fragmentation and indexing (tagmentation) was followed by target capture and enrichment. Reads were dual indexed by Nextera i7 and i5 primers. Libraries were prepared for subsequent cluster generation and sequencing on Illumina's NextSeq 500 platform (USA) using a 150-cycle output flow cell (V2 reagents). Samples were run at an envisioned depth of 100× per sample.

**Bioinformatics analysis.** Read mapping and variant calling were performed using Sentieon's DNAseq [48,49] FASTQ to VCF pipeline implemented on the DNAnexus cloud [USA]. Briefly, sequence reads were aligned to the GRCh38 reference genome using the BWA-MEM algorithm [50]. Duplicate reads were removed from further analysis. Base Quality Score Recalibration and indel realignment was performed on the mapped reads. Sentieon Haplotyper was exploited to call the variants and produce GVCF files. Sentieon GVCFtyper produced the final variant calling output as a VCF file. Only variants passing standard Sentieon filter criteria were used in the further analysis. Functional annotation of the variants was made using VarAFT v.2.151 [51]–a comprehensive annotation system for contextualizing variants and examining their functional consequences supported by multiple layers of disease phenotype-related databases.

**Variant filtering.** The first filtering step retained non-synonymous exonic variants or variants affecting splice donor/acceptor sites (±10nt) of canonical (longest) transcripts. Minor allele frequency (MAF) cut-off <1% was applied to 1000 Genomes, ExAC, and gnomAD genomic databases. Since only female samples were analyzed, autosomes and X chromosome variants were assessed identically. The second filtering step retained variants covered with at least 10 reads, with a variant allele frequency of at least 25%, and the following deleteriousness scores: Phred scaled CADD score [52] score >10, DANN [53] score >0.9, GERP [54] score >4, and excluded "benign" and "likely benign" variants of known clinical significance. Variants not assigned a particular score were also included in the further analyses.

**Variant classification, prioritization, and gene set enrichment analysis.** Each patient's gene variants retained after the second filtering step were pooled together to generate a single file containing rare deleterious variants. The list consisted of 1258 variants in total from 691 genes, 60 variants on average for each sample. Further, the pooled genetic variants were filtered using the three gene lists created by means of the systematic literature analysis (please refer to the 'Systematic literature analysis' section above for the detailed methodology).

The variants identified in genes from the first and second lists were considered to be of great interest and were consequently investigated more closely to discern the ones most likely to be contributive to the patients' phenotype. The pathogenicity of each variant from this list was assessed manually by three independent evaluators according to the ACMG guidelines [55] using MetaDome [56] and automatically using the online tool VarSome [57]. *In silico*-predicted mode of inheritance was assessed using the DOMINO tool (https://wwwfbm.unil.ch/domino/index.html). Gene expression patterns were assessed through a consensus dataset available at https://www.proteinatlas.org.

Variants located in splicing regions were analyzed using four splice prediction tools: SSF, MaxEntScan, NNSPLICE, and GeneSplicer [58–61] implemented on Alamut Visual v2.13. A variant was considered to have an effect on splicing if at least two of the four tools showed a >2% difference between the predicted splice scores of the wild-type and variant alleles, as described previously [62].

Further, to obtain unbiased information on pathway enrichments across the genes having rare and deleterious variants in our cohort, we annotated genes from the pooled list of variants (n = 1258) using the ConsensusPathDB interaction database [47] over-representation analysis and looked for 'Pathway-based sets' with a p-value cut-off of 0.01. As a background, we used the TruSight One gene list to exclude any bias from the target genes present in the kit.

## Results

### Systematic literature analysis

**Publication data.** We conducted a systematic literature analysis according to the PRISMA guidelines to cumulate published information on all genes assigned to cervical insufficiency and/ or playing a role in the biology of the cervix during normal/compromised pregnancy/parturition.

Out of 105 eligible studies selected for the gene extraction (S1 Fig; S2 Table), 51 were solely association studies with the majority focusing on the analysis of common genetic variants in a limited number of candidate genes (selected gene approach). Of those, five studies exploited array genotyping of 206–1536 single nucleotide polymorphisms (SNPs) in 9–190 genes. One study performed NGS analysis of 329 candidate genes, while only a single study employed a genome-wide association approach using genome-wide SNP arrays.

Four studies sought to find a direct genetic involvement in preterm delivery, cervical insufficiency, or PPROM However, all the studied patients were syndromic and not isolated cases of the aforementioned obstetrical complications. Thirty-eight studies were functional gene analyses performed on different sites of cervical biopsy, supracervical fetal membranes, and a few other tissues; 22 assessed differential RNA expression, while the remainder exploited other types of functional analysis. Again, the vast majority of the functional studies (31 in total) used selected gene approaches, with only seven employing the unbiased approach of genome-wide RNA microarray analysis. The rest of the studies used a combined analysis or a more sophisticated unbiased data analysis, e.g. case-parent triad design.

Overall, only eight genetic studies addressed cervical insufficiency primarily or in connection with preterm delivery, PPROM, or an associated genetic disorder. One was a functional study, five were association studies linking the condition with common genetic variants, and two were syndromic studies.

The majority of articles focused on preterm delivery (n = 66), whether primarily or in connection with PPROM. The rest of the studies were performed on healthy females at different times during pregnancy/parturition in order to study the physiological ripening of the uterine cervix and physiological pregnancy.

**Gene analysis: Genes linked to cervical insufficiency are mostly syndromic.** The gene extraction from the 105 selected publications resulted in 1455 entities with duplicates. Duplicate removal yielded 1181 genes. Our phenotype-based gene search using HPO terms (OMIM, GHR, https://hpo.jax.org) related to prematurity yielded 50 unique genes. The addition of syndromic genes resulted in the final list of 1222 unique genes (1509 with duplicates; S3 Table).

In total, 1024 genes (83.7%) were reported in the literature only once; the remainders were indexed at least twice. The most cited genes in relation to the genetics of the cervix were *IL6* (16 citations), *CXCL8* (12 citations), *IL1B* (10 citations), *TNF*, *PTGS2* (8 citations each), and *COL1A1*, *COL5A1* (7 citations each). Notably, *IL6* was mostly reported in publications focusing on selected gene approaches and was twice documented as being differentially expressed across seven genome-wide studies of RNA expression in cervical tissues.

Altogether, only 17 genes were primarily identified in relation to cervical insufficiency, with six being syndromic, i.e. *COL1A1* and *COL3A1* causing EDS; *FBN1* causing Marfan syndrome; *ZMPSTE24* and *LMNA* causing restrictive dermopathy; and *MATR3* causing myopathy. *COL3A1* was the only gene with an established gene-phenotype role as shown through HPO term 'Cervical insufficiency' (HP:0030009) along with 'Premature delivery because of cervical insufficiency or membrane fragility' (HP:0005267), 'Uterine rupture' (HP:0100718), and 'Uterine prolapse' (HP:0000139), and is known to cause EDS, vascular type (OMIM:130050).

The remaining genes were studied in association with cervical insufficiency using selected gene approaches. Five genes had no association (*ADRB2*, *IL1A*, *IL6R*, *LTA*, and *TNF*) as shown

**Table 2. Genes primarily linked to cervical insufficiency (first list of genes).**

| Gene | Associations from the literature and additional searches* |
|---|---|
| *COL1A1* | Ehlers-Danlos syndrome; Cervical insufficiency; Preterm delivery; PPROM; Physiological ripening of the uterine cervix; Physiological pregnancy |
| *COL3A1* | Ehlers-Danlos Syndrome; Cervical insufficiency HP:0030009/ Premature delivery because of cervical insufficiency or membrane fragility HP:0005267/ Uterine rupture HP:0100718/ Uterine prolapse HP:0000139; PPROM; Preterm delivery; Physiological ripening of the uterine cervix; Physiological pregnancy; Premature uterine contractions |
| *FBN1* | Marfan syndrome; Cervical insufficiency; PPROM; Premature uterine contractions |
| *HIF1A* | Cervical insufficiency; Physiological ripening of the uterine cervix; Physiological pregnancy |
| *IL10* | Cervical insufficiency; Preterm delivery |
| *IL1B* | Cervical insufficiency; Preterm delivery; Physiological ripening of the uterine cervix; Physiological pregnancy |
| *IL6* | Cervical insufficiency; Preterm delivery; Physiological ripening of the uterine cervix; Physiological pregnancy |
| *LMNA* | Restrictive Dermopathy; Premature delivery because of cervical insufficiency or membrane fragility HP:0005267; Premature rupture of membranes HP:0001788; |
| *MATR3* | Myopathy due to MATR3 mutations; Cervical insufficiency |
| *MBL2* | Cervical insufficiency; Preterm delivery |
| *TGFB1* | Cervical insufficiency; Preterm delivery; Physiological ripening of the uterine cervix; Physiological pregnancy |
| *ZMPSTE24* | Restrictive Dermopathy; Premature delivery because of cervical insufficiency or membrane fragility HP:0005267; PPROM; Preterm delivery |

*HPO term indicated if reported in https://hpo.jax.org.

by case-control studies [27,63,64]. They were therefore excluded from the first list of genes primarily linked to cervical insufficiency (n = 12; summarized in Table 2) composed from the data of the literature analysis and additional gene searches. Lastly, only one functional study was conducted to analyze *HIF1A* gene expression in the amniotic fluid from patients with isolated cervical insufficiency and indicated cerclage [65].

Based on the previously described clearly distinguishable clinical pattern of cervical insufficiency from classical idiopathic preterm labor and its relatedness to PPROM through the role of connective tissue–particularly the role of collagens–we composed a second list of genes (genes potentially linked to cervical insufficiency; S4 Table) comprising 91 entities also containing syndromic genes identified through the HPO/OMIM searches.

The third list contained 812 genes having a function within the uterine cervix as shown by differential gene expression studies of the physiology of pregnancy, cervical ripening, and labor (S5 Table). Studies focusing solely on idiopathic preterm delivery (both functional and DNA analysis) were excluded.

A Venn diagram was constructed from the three lists [66] (S2 Fig). Certain genes occurred in more than one list, indicating multiple lines of evidence. The creation of all three lists can be replicated through S5 Table column E.

**Functional annotation of genes studied using genome-wide versus selected gene approaches.** Lastly, to assess the bias in the existing knowledge on the genetics of cervical functioning, we analyzed differences in the biological information of genes reported in studies using unbiased genome-wide approaches in comparison to selected gene approach studies. In total, 816 genes emanated from seven genome-wide studies (excluding genome-wide studies subjected to gene filters). They were all gene expression studies exploiting genome-wide expression arrays. Specifically, 64 genes (7.8%) were denoted as being differentially expressed in more than one study (with a maximum of five studies). Eighty-seven genes emerged from

53 studies exploiting selected gene approaches, with 27 of them occurring multiple times (with a maximum of six studies). There were 27 genes occurring in both selected gene approach and genome-wide approach studies.

The gene lists were annotated using the ConsensusPathDB interaction database [47] for Gene Ontology (GO) and pathway enrichments. Annotation showed a large overlap among the strongest entities (as indicated by the p-value; S6 Table, S7 Table) and were enriched for GO terms primarily related to extracellular matrix (ECM) organization (e.g. GO:0031012; GO:0005201; GO:0007155) and a variety of cellular responses including immune (GO:0006955).

The pathway analysis showed a high enrichment of immune-related pathways overlapping between both gene lists (a variety of interleukins). Separately, each list showed different pathways, but, again, they could be accommodated under common denominators related to elastic fiber/collagen formation as well as immunity (e.g. 'ECM proteoglycans'; 'Elastic fiber formation'; 'Collagen formation'; etc.).

### Patient NGS data analysis

Using Illumina's TruSight One NGS kit covering 4810 genes of known clinical significance, we sequenced the DNA from 21 patients presenting with isolated non-syndromic cervical insufficiency. The sequencing resulted in a median coverage depth of 135±38× of the target region, with 94.4% of target regions being covered at least 10 times and 89.7% being covered at least 20 times. Rare deleterious variants (filtered based on CADD, DANN, and GERP scores and known clinical significance; S8 Table) from all the patients were pooled and screened for the genetic variations in the three lists created from the literature analysis (Fig 2).

Twenty heterozygous variants found in 14 of our patients (67%) and the first and second lists of genes were subjected to a closer analysis as they were considered most likely to contribute to the patients' phenotype based on existing knowledge (Table 3). Nine patients had one variant, four patients had two variants, and one patient had three variants. Fourteen variants were found in 10 genes known to cause EDS, osteogenesis imperfecta, or Bethlem myopathy. According to manual classification following the ACMG guidelines, 14 variants were classified as variants of unknown significance (VUS, class 3) and two as likely pathogenic (class 4). The criteria were inapplicable to the remaining four variants (please see the full information about each variant in S9 Table).

**3rd List: Genes playing a role within the cervix**
(n=812; 326 in TruSight)
67 variants in our cohort

**2nd list: Genes potentially linked to cervical insufficiency**
(n=91; 65 in TruSight)
18 variants in our cohort

**1st List: Genes primarily linked to cervical insufficiency**
(n=12; All present in TruSight)
2 variants in our cohort

**Fig 2. Each circle represents one of the three gene lists generated in the study.** The smaller the list, the closer the association with cervical insufficiency. The number of genes covered in the TruSight NGS kit are mentioned, as well as the number of deleterious variants identified in our patients across each gene list.

**Table 3. Closer analysis of variants most likely contributing to the development of cervical insufficiency in our patients.**

| Sample | Gene | Genotype (effect on protein if known) | GnomAD v2.1.1. | CADD score | ACMG Manual (Criteria) | Comments | Contribution to the phenotype of cervical insufficiency |
|---|---|---|---|---|---|---|---|
| Case1 | MYO1F (rs200225777) | NM_012335.4:c.[2461G>A]; [= ] (NP_036467.2:p.Gly817Arg) | 0.000106 (30/280708) | 23.2 | Not applicable[a] (PP3; BP6) | Mechanism of the disease is unknown and no phenotype for the gene is known. Likely recessive type of inheritance. | Unlikely |
| Case2 | FKBP14 (rs542254849) | NM_017946:c.[496_498del]; [= ] (NP_060416.1:p.Lys166del) | 0.001070 (279/260856) | 15 | VUS (PP3; PM4; PM1) | Pathogenic variants in the gene cause EDS, which is associated with cervical insufficiency. Likely recessive type of inheritance. | Further investigation needed |
| Case3 | B4GALT7 (rs142476892) | NM_007255:c.[277C>T]; [= ] (NP_009186.1:p.His93Tyr) | 0.001347 (379/278326) | 26.7 | VUS (PP3; PM1) | Pathogenic variants in the gene cause EDS;VUS previously found in EDS patients; Likely recessive type of inheritance. | Further investigation needed |
|  | COL1A2 | NM_000089:c.[1808C>T]; [= ] (NP_000080.2:p.Thr603Ile) | 0 | 17 | VUS (PP3; PM2) | Pathogenic variants in the gene cause EDS, which is associated with cervical insufficiency and PPROM; Strongest expression within the cervix. Likely dominant type of inheritance. | Further investigation needed |
| Case4 | COL1A1 (rs778463556) | NM_000088:c.[1663C>T]; [= ] (NP_000079.2:p.Pro555Ser) | 0.000021 (6/282304) | 24.6 | LP (PP3; PM5; PP2; PM2) | Pathogenic variants in the gene cause EDS; Strong expression within cervix; Likely dominant type of inheritance. | Further investigation needed |
| Case5 | COL12A1 (rs201988277) | NM_004370.6:[c.7853C>T]; [= ] (NP_004361.3:p.Thr1454Met) | 0.000311 (74/237906) | 25.2 | VUS (PP3) | Pathogenic variants in the gene cause Ehlers-Danlos/Bethlem-like myopathy overlap syndrome associated with both connective tissue abnormalities and muscle weakness. Likely dominant type of inheritance. | Further investigation needed |
|  | COL1A1 (rs537060488) | NM_000088:c.[529G>A]; [= ] (NP_000079.2:p.Val177Met) | 0.000014 (4/281442) | 23.2 | VUS (PP2) | Pathogenic variants in the gene cause EDS; Strong expression within cervix; Likely dominant type of inheritance. | Further investigation needed |
| Case6 | CHST14 (rs144629123) | NM_130468:c.[635T>C]; [= ] (NP_569735.1:p.Val212Ala) | 0.000513 (145/282534) | 25 | VUS (PP3; PM1; BS2) | Strong expression within cervix. Gene is associated with EDS. Likely dominant type of inheritance. | Further investigation needed |
|  | GK (rs371481560) | NM_000167:c.[989G>A]; [= ] (NP_976325.1:p.Arg330His) | 0.00001 (4/204700) | 25.6 | VUS (PP3; PM1) | Phenotype of the disease associated with GK gene does not overlap with the phenotype of interest; Poor expression with within cervix. X linked recessive. | Unlikely |
| Case7 | MYO1F (rs761308378) | NM_001348355:c.[2270G>A]; [= ] (NP_036467.2:p.Arg757Gln) | 0.000224 (63/280790) | 34 | Not applicable[a] (PP3) | Mechanism of the disease is unknown and no phenotype for the gene is known. Likely recessive type of inheritance. | Unlikely |
|  | COL4A3 (rs765655100) | NM_000091.4:c.[5010_*14del]; [= ] (NP_000082.2:p. His1670_Ter1671delinsXaa) | 0.000064 (16/249376) | 40 | LP (PP3; PM1; PM4; PM2) | Localized at the end of the gene (loss of stop-codon); Poor expression within cervix; Dominant or recessive. | Unlikely |
| Case8 | TNXB | NM_001365276:c.[3793G>A]; [= ] (NP_001352205.1:p.Gly1265Arg) | 0 | 25.9 | VUS (PP3; PM2; BP1) | Gene is associated with EDS hypermobile type; Likely recessive type of inheritance. | Further investigation needed |

(*Continued*)

**Table 3.** (Continued)

| Sample | Gene | Genotype (effect on protein if known) | GnomAD v2.1.1. | CADD score | ACMG Manual (Criteria) | Comments | Contribution to the phenotype of cervical insufficiency |
|---|---|---|---|---|---|---|---|
| Case9 | *B4GALT7* (rs142476892) | NM_007255:c.[277C>T]; [= ] (NP_009186.1:p.His93Tyr) | 0.001347 (375/ 278326) | 26.7 | VUS (PP3; PM1) | Pathogenic variants in the gene cause EDS; VUS previously found in EDS patients; Likely recessive type of inheritance. | Further investigation needed |
| | *TNXB* (rs141190850) | NM_001365276:c.[2030A>G]; [= ] (NP_061978.6:p.Glu677Gly) | 0.002140 (583/ 272438) | 23.1 | VUS (PP3; BP1) | Gene is associated with EDS hypermobile type; Likely recessive type of inheritance. | Further investigation needed |
| Case10 | *PLOD1* (rs772861343) | NM_000302:c.[475G>A]; [= ] (NP_001303249.1:p.Gly159Ser) | 0.000035 (9/ 251374) | 34 | VUS (PP3; PM2) | Pathogenic variants in the gene cause EDS; VUS previously found in EDS patients; Strong expression within the cervix; Likely recessive type of inheritance. | Further investigation needed |
| Case11 | *P3H1* (rs371232413) | NM_001146289:c.[1720+4G>A]; [= ] | 0.000213 (59/ 276822) | 5.4 | VUS | Pathogenic variants in the gene known to cause Osteogenesis imperfecta, which was clinically associated with cervical insufficiency; Effect on splicing is not clear. | Further investigation needed |
| Case12 | *P3H1* (rs371232413) | NM_001146289:c.[1720+4G>A]; [= ] | 0.000213 (59/ 276822) | 5.4 | VUS | Pathogenic variants in the gene known to cause Osteogenesis imperfecta, which was clinically associated with cervical insufficiency; Effect on splicing is no clear; Likely recessive type of inheritance. | Further investigation needed |
| | *C1S* (rs148105120) | NM_001734:c.[100A>G]; [= ] (NP_001725.1:p.Ser34Gly) | 0.000438 (124/ 282870) | 22.4 | VUS (PP3; BS1) | Associated with EDS, but gene-disease association is dubious (one missense variant reported in publication); Likely recessive type of inheritance. | Unlikely |
| | *MYO1F* (rs747756979) | NM_001348355:c.[1170+4C>T]; [= ] | 0.000039 (11/280872) | 0 | Not applicable[a] (BP4) | Mechanism of the disease is unknown and no phenotype for the gene is known. Variant does not have a consistent impact on the splice site; Likely recessive type of inheritance. | Unlikely |
| Case13 | *ADRB2* (rs753894727) | NM_000024:c.[1072G>C]; [= ] (NP_000015.1:p.Gly358Arg) | 0.000074 (28/282828) | 12.5 | Not applicable[a] (BP1; BP4) | Gene is not associated with phenotype; Study, from which information is extracted, looked only for SNP and did not find any association with cervical insufficiency; Likely dominant inheritance. | Unlikely |

[a] Not applicable—if mechanism of disease is not known, or phenotype is not known; VUS—variant of unknown significance, LP—likely pathogenic.

Two variants were located at non-canonical splicing sites (>2nt from exon/intron junction). Based on splicing predictions, variant *P3H1*:c.1720+4G>A (found in two patients) strengthens the existing donor site and may have a mild effect on the creation of a new acceptor site at position c.1720+23. The variant *MYO1F*:c.1170+4C>T does not have a consistent impact on the splice site.

Ultimately, based on a comprehensive curation of the variants' pathogenicity, including known gene-disease/gene-phenotype associations, gene expression patterns within cervical tissues (S3 Fig), and mechanisms of diseases of particular genes, etc. (all criteria used for the

curation can be found in S9 Table), we assigned a likelihood for contribution of the variant to the patient's phenotype (last column in Table 3). A variant was *unlikely contributing* (n = 7) if classified as benign/likely benign according to the manual pathogenicity curation, did not show any or poor expression within the cervix, or known gene-disease/gene-phenotype associations did not correspond to the phenotype of interest. A variant *needs further investigation* (n = 13) if it showed a theoretical potential to increase susceptibility to the development of the phenotype of interest based on the criteria assessed, but more data are required to declare the variant as definitively contributive to the development of cervical insufficiency.

**Gene pathway enrichment analysis.** To determine whether the genes having rare deleterious variants identified in our highly selective patient cohort exhibited any phenotype-relevant pathway enrichment, we annotated all the genes (n = 694) using the ConsensusPathDB interaction database [47] with the TruSight One gene list (n = 4810) as background. As illustrated by the 20 most significant entities (Table 4), the analysis revealed a high overrepresentation of pathways related to tissue mechanical and biomechanical properties (collagens and proteoglycans, integrins). There was not only high enrichment of ECM pathways, but also of cell to ECM communication (e.g. hemidesmosomes, focal adhesion) and basal membrane components (laminins). Moreover, a number of the pathways identified here matched ones shown to be enriched with genes studied in relation to the genetics of the cervix as identified from our literature search (marked with an asterix, Table 4).

## Discussion

An evidence-based, effective system of pregnancy and maternity care identifying risks and localizing problems in a timely manner is one of the fundamental public health elements for maintaining the well-being and demography of a population. However, obstetrics and gynecology practice is associated with a plethora of different complications, thus placing a burden on the healthcare and socio-economic systems and is a traumatizing experience for the patient. A distinct risk factor for pregnancy loss and preterm delivery is cervical insufficiency. Currently, cervical insufficiency is clinically distinguishable only in the ongoing pregnancy or based on a patient's anamnesis.

A strong genetic component is expected in certain cases of preterm delivery and cervical insufficiency [21,22]. However, despite genomic advancements, we know surprisingly little about the genetics of prematurity. Indeed, until recently, there was not a single gene unequivocally linked to the specific phenotypes associated with cervical insufficiency or PPROM and PTB.

In this work, we aimed to comprehensively explicate the existing genetic studies on prematurity with cervical insufficiency as the focal point. To accomplish this, we performed a systematic literature analysis followed by a gene extraction and data analysis. We wanted to evaluate whether there is a bias in our understanding of the genetics of the cervix and estimate how many genes can be reliably linked to cervical insufficiency and explore their possible roles. We subsequently applied the obtained knowledge from the literature to the analysis of NGS data of 21 patients with an anamnesis of isolated cervical insufficiency. As evidenced in previously conducted research, it is still not clear whether the mother or the preterm-delivered infant should be considered the proband, and as a result, which individual's DNA should be examined [18]. Based on the available evidence, we hypothesized that issues relating to the cervix during pregnancy are most likely to be dictated by the maternal genome. Therefore, we analyzed the mother's genomic DNA.

### Collagenopathic nature of cervical insufficiency

Our literature analysis revealed that at present there are only eight studies directly addressing the link between genetics and cervical insufficiency, and 12 genes are primarily linked to the

**Table 4. Pathway enrichment analysis of genes having rare deleterious variants in patients with cervical insufficiency.**

| Pathway name | Pathway set size (Number in the background gene list) | Genes contained in the analyzed list | p-value | q-value | Pathway Source |
|---|---|---|---|---|---|
| **Collagen formation*** | 92(59) | 24 (40.7%) | 3.30E-07 | 4.71E-04 | Reactome |
| **ECM-receptor interaction—Homo sapiens (human)** | 82(60) | 24 (40.0%) | 4.80E-07 | 4.71E-04 | KEGG |
| **Extracellular matrix organization*** | 294(180) | 47 (26.1%) | 7.21E-06 | 4.33E-03 | Reactome |
| **Collagen biosynthesis and modifying enzymes*** | 68(40) | 17 (42.5%) | 8.82E-06 | 4.33E-03 | Reactome |
| **Type I hemidesmosome assembly** | 9(9) | 7 (77.8%) | 2.76E-05 | 9.10E-03 | Reactome |
| **Assembly of collagen fibrils and other multimeric structures** | 48(39) | 16 (41.0%) | 2.78E-05 | 9.10E-03 | Reactome |
| **Laminin interactions** | 23(19) | 10 (52.6%) | 7.26E-05 | 2.04E-02 | Reactome |
| Axon guidance | 358(160) | 40 (25.0%) | 1.02E-04 | 2.51E-02 | Reactome |
| **Collagen chain trimerization** | 44(28) | 12 (42.9%) | 1.73E-04 | 3.78E-02 | Reactome |
| Human papillomavirus infection—Homo sapiens (human) | 339(173) | 41 (23.7%) | 2.95E-04 | 5.79E-02 | KEGG |
| **Integrin*** | 124(75) | 22 (29.3%) | 3.71E-04 | 6.63E-02 | INOH |
| **Non-integrin membrane-ECM interactions*** | 42(31) | 12 (38.7%) | 5.39E-04 | 8.83E-02 | Reactome |
| **ECM proteoglycans*** | 57(41) | 14 (34.1%) | 8.36E-04 | 1.24E-01 | Reactome |
| **Beta1 integrin cell surface interactions*** | 66(55) | 17 (30.9%) | 8.85E-04 | 1.24E-01 | PID |
| Protein-protein interactions at synapses | 88(33) | 12 (36.4%) | 1.04E-03 | 1.36E-01 | Reactome |
| **Focal adhesion—Homo sapiens (human)** | 199(113) | 28 (24.8%) | 1.30E-03 | 1.59E-01 | KEGG |
| **Alpha6 beta4 integrin-ligand interactions** | 11(11) | 6 (54.5%) | 1.77E-03 | 2.05E-01 | PID |
| Developmental Biology | 620(262) | 53 (20.2%) | 2.40E-03 | 2.55E-01 | Reactome |
| Interaction between L1 and Ankyrins | 29(23) | 9 (39.1%) | 2.47E-03 | 2.55E-01 | Reactome |
| *Myo*-inositol *de novo* biosynthesis | 4(3) | 3 (100.0%) | 2.70E-03 | 2.65E-01 | Human Cyc |

Pathways in bold are related to tissue mechanical and biomechanical properties. Pathways marked with an asterix were found to be enriched with genes studied in relation to the genetics of the cervix as identified from our literature search.

condition. A few of them are known to cause syndromic forms of cervical insufficiency associated with collagen disorders such as EDS, Marfan syndrome, restrictive dermopathy, and myopathy due to *MATR3* mutations. Notably, no studies have been conducted to identify direct genetic implications in the non-syndromic form of cervical insufficiency, highlighting the insufficient knowledge on the genetics of (patho)physiological cervical remodeling during pregnancy.

All the genes retrieved in this study were classified into three lists according to their relation to the question being posed. The second list contained genes (n = 91) procured based on

knowledge of the importance of collagen for proper cervical morphophysiology. Not surprisingly, 32 of the genes in the list (approximately one third) have been shown to associate with collagenopathy syndromes. Consequently, we anticipated that a large number of these genes also play a crucial role in the development of cervical insufficiency. Therefore, we comprehensively evaluated our patients' NGS variants identified in genes classified in the first and second gene lists by adhering to the robust framework for variant interpretation in the research setting in order to identify ones potentially contributive to the development of cervical insufficiency.

Currently, only one gene–*COL3A1*–has an established gene-disease relationship with cervical insufficiency as shown through HPO term association (HP:0030009). This gene is also linked to 'Premature delivery because of cervical insufficiency or membrane fragility' (HP:0005267), as are the genes *ZMPSTE24* and *LMNA*. Further, *COL5A1* is the only gene associated with 'Premature birth following premature rupture of fetal membranes' (HP:0005100) and 'Premature rupture of membranes' (HP:0001788), with a few other genes (*PLOD1*, *ADAMTS2*, *SERPINH1*, *ZMPSTE24*, *LMNA*, and *ATP6V0A2*) associated with PPROM alone. Notably, we identified only one VUS in these genes in our patient cohort. The patient carrying the *PLOD1*:c.475G>A variant had two preterm deliveries, a cervical length of 1.8 cm as identified during her last pregnancy, and no history of PPROM. However, the variant's contribution to the patient's phenotype needs to be investigated further as the suggested inheritance is autosomal recessive and the described gene-phenotype correlation is more severe.

Another collagen gene–*COL1A1*–has already been implicated in the development of cervical insufficiency from the data of large case-control studies [22,27] showing a positive genetic association (OR >3) and pathogenic variants in which known to cause EDS similarly to *COL1A2* gene. We believe both genes, *COL1A1* and *COL1A2*, are good candidate genes for involvement in the development of isolated cervical insufficiency. Nonetheless, the variants identified in our cohort, *COL1A1*:c.1663C>T, *COL1A1*:c.529G>A, and *COL1A2*:c.1808C>T, require more in-depth investigation and replication to be reliably assigned as causative.

Further, we identified the same variant in the *B4GALT7* gene in two of our patients. This gene is known to cause EDS, autosomal recessive type. Similarly, two patients were found to carry the same variant in the *P3H1* gene causing autosomal recessive osteogenesis imperfecta. Notably, all the other likely pathogenic variants and VUSs, excluding ones classified as unlikely contributive, were identified in genes in which pathogenic variants/mutations might lead to the development of certain types of collagenopathy. Such a situation raises the question: is it the case that patients carrying variants likely causing certain connective tissue disorders, do not exhibit any other collagenopathic features? Indeed, no one, to our knowledge, has evaluated cervical insufficiency as an expression point in a phenotypic continuum of collagenopathies when no other symptoms or subtle symptoms are apparent–and neither did we. Therefore, our findings allow us to hypothesize further that cervical insufficiency could be expressed as one of the mild forms of collagenopathy, a condition which is known to range from mildly loose joints to life-threatening complications such as aortal rupture. In order to substantiate this, a rigorous phenotyping following a custom-developed assessment protocol would have to be conducted as at present the only existing and validated scale used worldwide for collagenopathies is the Beighton joint hypermobility score [67], which does not address any other disease-related symptoms apart from hypermobility of joints.

In general, reports linking obstetrical complications with EDS date back to the 1990s with descriptions of patients with hypermobile joints, kyphoscoliosis, and hyper elastic skin having cervical insufficiency and PPROM [68,69]. Additional complications in cases of EDS might include scoliosis (causing problems with anaesthesia), atonic uterus, vaginal and/or perineal tearing, pelvic organ prolapse, symphysiolysis, abdominal herniation, wound dehiscence, severe varicosities, and postpartum hemorrhage. Maternal mortality risk is heightened due to

uterine rupture or rupture of large vessels [70,71]. It is also known that the coincidence of Marfan syndrome and pregnancy means a high risk for mother and child as it might be complicated by PPROM, premature uterine contractions, and cervical insufficiency [72]. Therefore, knowledge that the patient is suspected of having or indeed has a connective tissue disorder or a tendency towards connective tissue laxity (perhaps only in certain scenarios) would provide a unique opportunity for the multidisciplinary team members to more effectively support pregnant women through increased understanding and awareness [73].

## Main pathways involved in cervical functioning

Candidate genes from a variety of pathways have been shown to be important in prematurity: hemostasis and coagulation, local inflammation, collagen metabolism, and matrix degradation in PPROM [37]; focal adhesion, cell communication, and ECM receptor interaction in spontaneous PTB [74]. No functional gene studies with subsequent pathway enrichment analysis have been performed in relation to cervical insufficiency.

In this study, we wanted to evaluate whether there is a bias in the cervix-related genes studied exploiting selected gene approaches in contrast to unbiased genome-wide approach studies. Our pathway and GO enrichment analysis of both groups did not identify a large difference in the overrepresented GO terms/enriched biological pathways and showed a certain overlap with other phenotypes in prematurity. The genes studied in relation to cervical patho(physiology) during pregnancy/parturition were found to be mostly enriched for the two main functional categories–immunity/inflammation and connective tissue remodeling. This indicates that the genes chosen for the selected gene approach studies follow the existing understanding of the most well-known biological pathways in cervical remodeling.

Importantly, the process of parturition at both term and preterm is consistently associated with the induction of many proinflammatory mediators, suggesting that these components are central for the parturition cascade in humans [75]. In turn, the inflammatory infiltrate *per se* activates fibrinolysis [68]. As a consequence, true (transcript)omic signature comprising less known pathways of the specific phenotype might simply be masked by the massive expression of (pro)inflammatory agents followed by fibrinolysis. It is worth mentioning that the first two phases of cervical remodeling, namely, softening and ripening, are not dependent on inflammatory processes [76], and so may dictate the timing of sampling in at least physiological pregnancy/parturition studies.

Despite study limitation of relatively small sample cohort, our unbiased *in silico* pathway enrichment analysis of genes having rare deleterious variants in our patiens–specifically selected for isolated non-syndromic cervical insufficiency–identified an increased variant burden in genes involved in collagen and/or ECM production. These results not only strengthen our target gene variation findings that pathways involved in collagen biosynthesis play a major role in cervical insufficiency, but also imply that cell-ECM communication pathways, in which molecules such as integrins, laminins, keratins, and fibronectins participate, might be involved in the development of cervical insufficiency. For example, one of the enriched pathways was 'Type I hemidesmosome assembly'. Hemidesmosomes play a critical role in the maintenance of tissue integrity and are highly dynamic structures capable of disassembling quickly during cell division, differentiation, or migration [77]. Desmosomes present in the uterine cervix [78]–the functioning of which might be compromised due to genetic variations–can also affect the integrity of the cervix. There is currently insufficient information on the genes of these pathways to reliably implicate them in the pathology of the cervix. Therefore, this could be a focus for future studies.

## Overview and future perspectives

The findings of our systematic overview of the genes related to (patho)physiological cervical remodeling are a further step towards unraveling the complex genomics of prematurity. Specifically, the obtained data should help to fill in the gaps in our knowledge about cervical insufficiency, as there is still controversy surrounding this condition's development and treatment due to a largely unclear pathophysiology [22]. In the future, findings from molecular-based studies could potentially be translated into outcome changes for women at risk, as they may lead to the discovery of a particular metabolite's deficit and consequently the development of screening tests. Since the biophysical properties of the cervix are mainly determined by collagen content [41], perhaps the basis of future screening tests lies in the observed collagen changes during cervical remodeling. It has been demonstrated that women with cervical insufficiency exhibited a markedly lower median cervical hydroxyproline (the most abundant amino acid in the collagen molecule) concentration, high collagen extractabilities and collagenolytic activities, and their biomechanically tested biopsy specimens had low strength and high extensibility [79], even in the non-pregnant state [80]. Unfortunately, our patients did not undergo such testing.

We have demonstrated for the first time that rare pathogenic allelic variants leading to collagenopathies might be responsible for the increased susceptibility of the development of isolated cervical insufficiency in non-syndromic patients. Nonetheless, the genetic landscape summarized within the scope of this work points to a wider genetic heterogeneity of the condition. At present, the majority of genes (particularly those listed in the third list of genes; S5 Table) cannot with certainty be implicated in the development of cervical malfunctioning. The third gene list contains data from functional studies and encompasses genes shown to be differentially expressed within the cervix during physiological pregnancy in healthy females. With the current knowledge, it is difficult to interpret the contribution of the deleterious variants identified in those genes (n = 67). However, the genes are still likely to play an important role in cervical functioning and/or preterm delivery and further evidence might emerge as knowledge increases over time. Further studies with larger patient cohorts and perhaps better designs–specifically, precise tissue selection in combination with the timing of sampling during physiological/compromised pregnancy/labor, and most importantly precise phenotyping of patients–are necessary before any conclusions regarding thorough genetics of cervical insufficiency and clinical applicability of any genetic testing can be drafted.

A possible limitation of our extensive literature analysis could be an issue with multiple terms associated with cervical insufficiency. For example, the term PreCocious Cervical Ripening (PCCR) was initially coined by Papiernik *et al.* in 1986 [81] and was proposed as more appropriate and less confusing [82,83] than others such as cervical weakness/incompetence, (istmo)cervical insufficiency, premature cervical shortening/remodeling/failure, or failing cervix. Although all of them are used, we encourage usage of 'cervical insufficiency' since this is adopted by the American College of Obstetricians and Gynecologists, HPO, and MeSH. We tried to include all known terms of the condition in our literature search, but specific forms may have been omitted.

Another limitation of our study is the usage of a limited NGS panel (4810 genes) to study the DNA of our patients. Nevertheless, we were able to analyze all the genes primarily linked to cervical insufficiency as revealed by our literature analysis results. Having a relatively small patient cohort, we wanted to elucidate the particular roles of rare pathogenic variants since they are thought to have greater effects on the development of complex human diseases in comparison to common genetic variants, testing of which exploits the idea 'common disease–common variation' [84] and demands enormous patient cohorts. Ultimately, by means of the

target gene variation analysis and pathway enrichment analysis, we were able to demonstrate that the development of isolated cervical insufficiency is likely influenced by rare variations in genes involved in ECM/collagen production and synthesis.

Despite the distinctive phenotypical pattern of cervical insufficiency in contrast to idiopathic preterm delivery, it still remains a multifactorial condition, development of which depends on a number of factors. Therefore, no marker will be 100% sensitive or specific. From the genetic point of view, the most likely scenario is that a combination of both multiple rare and common variants in a number of genes contributes to disease development risk.

However, before the era of genetic testing enters obstetrics and gynecology, there are measures that can be introduced now into clinical practice. For instance, routine cervical length screening is not always performed on low risk women. This paucity of screening may lead to a clinically unrecognized short cervix being missed and ultimately preterm labor [13]. Additionally, a number of EDS types are poorly recognized, with symptoms/complaints of generalized joint hypermobility and/or chronic musculoskeletal pain being wrongly attributed to rheumatologic disorders [68] or other non-specific conditions. Therefore, awareness regarding the nature of cervical insufficiency needs to be raised among obstetrical and gynecological teams to, at best, avoid complications or, at least, successfully manage them.

## Conclusions

Isolated cervical insufficiency is a distinct phenotype in prematurity with a heterogeneous etiology. Our current understanding of the genetic landscape of the (patho)biology of the cervix is incomplete. One of the causes of non-syndromic cervical insufficiency may be associated with pathogenic variants in genes involved in collagen synthesis and production, allelic variants in which are known to cause connective tissue disorders. The notion that cervical insufficiency is an expression point in a phenotypic continuum of collagenopathies should be investigated further using multiple approaches.

## Supporting information

**S1 Table. Full text articles assessed for eligibility according to the PRISMA guidelines.**
(XLS)

**S2 Table. Publication analysis.**
(XLS)

**S3 Table. Full list of genes.** All genes with duplicates obtained from a literature analysis and additional gene searches.
(XLS)

**S4 Table. The second list of genes.** Containing genes selected for association with cervical insufficiency, PPROM, or other connective tissue-related issues.
(XLS)

**S5 Table. The third list of genes.** Containing genes having a function within the cervix as identified from functional studies of physiological/compromised pregnancy/parturition.
(XLS)

**S6 Table. GO analysis of genes identified from genome-wide and selected gene approaches.**
(XLS)

**S7 Table. Pathway analysis of genes identified from genome-wide and selected gene approaches.**
(XLS)

**S8 Table. Sequence variants retained after each filtering step.**
(XLS)

**S9 Table. Full analysis of selected gene variants of interest.**
(XLS)

**S1 Fig. Genetic studies included in the study.** All studies included for the gene selection (left) and ones focusing primarily on cervical insufficiency (right).
(TIF)

**S2 Fig. Venn diagram of the three gene lists created from publication data and additional gene searches.**
(TIF)

**S3 Fig. Gene expression in the uterine cervix.** Genes found to be mutated in our patient cohort as shown through the 1st and 2nd gene list analysis. Data obtained through an RNA expression dataset available at https://www.proteinatlas.org. In our cohort, none of the rare or pathogenic variants were found in the COL3A1 gene; however, it is included as it is the only gene unequivocally linked to cervical insufficiency.
(TIF)

**S1 Data.**
(DOC)

## Acknowledgments

We would like to express our special thanks of gratitude to Riga Stradins University librarian Natalija Kislakova for her the help in the literature search.

## Author Contributions

**Conceptualization:** Ludmila Volozonoka, Dace Rezeberga, Linda Gailite, Anna Miskova.

**Data curation:** Ludmila Volozonoka, Dmitrijs Rots, Anna Kornete, Linda Gailite, Anna Miskova.

**Formal analysis:** Dmitrijs Rots.

**Funding acquisition:** Inga Kempa, Linda Gailite.

**Methodology:** Ludmila Volozonoka, Dmitrijs Rots, Anna Kornete, Linda Gailite, Anna Miskova.

**Project administration:** Inga Kempa, Linda Gailite, Anna Miskova.

**Resources:** Inga Kempa, Dace Rezeberga, Linda Gailite, Anna Miskova.

**Software:** Ludmila Volozonoka, Dmitrijs Rots.

**Supervision:** Inga Kempa, Dace Rezeberga, Linda Gailite, Anna Miskova.

**Visualization:** Ludmila Volozonoka.

**Writing – original draft:** Ludmila Volozonoka.

**Writing – review & editing:** Dmitrijs Rots, Inga Kempa, Anna Kornete, Dace Rezeberga, Linda Gailite, Anna Miskova.

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
