## [Decision Letter · Decision Letter 0]

2 Mar 2020

PONE-D-20-02423

Genetic landscape of preterm birth due to cervical insufficiency: comprehensive gene analysis and patient next-generation sequencing data interpretation

PLOS ONE

Dear Mrs Volozonoka,

Thank you for submitting your manuscript to PLOS ONE. After careful consideration, we feel that it has merit but does not fully meet PLOS ONE’s publication criteria as it currently stands. Therefore, we invite you to submit a revised version of the manuscript that addresses the points raised during the review process.

The reviewers found merit in this manuscript but raised some points. Please address them carefully and also discuss the strength and weakness of this study as the number of patients is low.

We would appreciate receiving your revised manuscript by Apr 11 2020 11:59PM. To enhance the reproducibility of your results, we recommend that if applicable you deposit your laboratory protocols in protocols.io, where a protocol can be assigned its own identifier (DOI) such that it can be cited independently in the future. For instructions see: http://journals.plos.org/plosone/s/submission-guidelines#loc-laboratory-protocols

We look forward to receiving your revised manuscript.

Kind regards,

Obul Reddy Bandapalli, MSc, PhD

Academic Editor

PLOS ONE

Journal Requirements:

Reviewers' comments:

Reviewer's Responses to Questions

**Comments to the Author**

1. Is the manuscript technically sound, and do the data support the conclusions?

Reviewer #1: Yes

Reviewer #2: Yes

2. Has the statistical analysis been performed appropriately and rigorously? 

Reviewer #1: I Don't Know

Reviewer #2: Yes

3. Have the authors made all data underlying the findings in their manuscript fully available?

Reviewer #1: Yes

Reviewer #2: Yes

4. Is the manuscript presented in an intelligible fashion and written in standard English?

Reviewer #1: Yes

Reviewer #2: Yes

5. Review Comments to the Author

Reviewer #1: In this research article by Volozonoka et al. Genetic basis of cervical insufficiency is investigated by performing NGS of genes implicated in cervical functioning in patients with cervical insufficiency. The manuscript is overall well written and the the proposed hypothesis of cervical insufficiency being a subtle form of collagenopathy is well supported. It woulb then interesting to add to the report information on the familial history of the analyzed patients, highlighting whether there were additional cases of cervical insufficiency or collagenopathy-related disorders, that would strenghten the idea of a common genetic basis.

Reviewer #2: Authors performed a genetic study on preterm birth due to cervical insufficiency with the support of systematic literature review and NGS analysis of 21 patients based on these review results.

Systematic review was done in detail and the lists were provided as supplementary files.

Although the number of patients is too low to make any significant conclusion about cervical insufficiency based on findings, especially for rare variants, I believe this study still be publishable for the purpose of using it as a guide and summary of reviewing the published papers associated with cervical insufficiency. For that purpose, especially for the genetic findings the tone of the paper couldn’t/shouldn’t be too assertive. Authors did already a good job for being NOT assertive for the findings on 21 samples.

I have some additional comments and suggestions;

In the manuscripts authors indicated that (line number 399-401)“The sequencing resulted in a median coverage depth of 135±38× of the target 400 region, with 94.4% of target regions being covered at least 10 times and 89.7% being covered 401 at least 20 times.”

What about the depth of coverage for rare deleterious variants, if they can mention this in that section, it would be very useful.

Line 60, "Sention", it would be useful to provide a reference in parenthesis as it is citing the kits. Example:(USA, CA)

Line 62, Please cite BWA-MEM algorithm. Although it is very well-known algorithm in the field, for the readers who are not familiar, it is needed to be cited.

Line 75, CADD score threshold was not mentioned, usually it is 15 and higher for pathogenic variants. It would be better to write the threshold, since thresholds for other filters were indicated.

Line 76, PHRED Score >10 is too low, which means probability of 1 in 10 of incorrect base call, usually 30 is and higher is preferred. I would recommend strongly to adjust it to 30 for this.

6. PLOS authors have the option to publish the peer review history of their article (what does this mean?). If published, this will include your full peer review and any attached files.

Reviewer #1: No

Reviewer #2: No

---

## [Author Response · Author response to Decision Letter 0]

4 Mar 2020

Dear Obul Reddy Bandapalli,

Dear reviewers,

We are grateful for your careful and thorough review of our manuscript, your comments and advices. We have in details addressed all of the points and made appropriate changes to the manuscript as well as provided answers below.

We would like to emphasize that despite study limitation of relatively small sample cohort, the strength of it is rigorous approach to the data analysis. First of all, to have a solid start and a priori knowledge of the question we performed systematic gene analysis and comprehensively exploited this information to analyze patients’ NGS data. Importantly, our findings made through target gene analysis was strengthened by the unbiased in silico pathway enrichment analysis. We have discussed this issue in the lines 577-584 (here and further lines are provided from the manuscript version with track changes) of the manuscript and also made a few changes to that. We think the results obtained in the study can be a good preliminary data point for the further investigations.

Reviewer #1: In this research article by Volozonoka et al. Genetic basis of cervical insufficiency is investigated by performing NGS of genes implicated in cervical functioning in patients with cervical insufficiency. The manuscript is overall well written and the the proposed hypothesis of cervical insufficiency being a subtle form of collagenopathy is well supported. It woulb then interesting to add to the report information on the familial history of the analyzed patients, highlighting whether there were additional cases of cervical insufficiency or collagenopathy-related disorders, that would strenghten the idea of a common genetic basis.

Answer: Thank you very much for your comment and the important notion regarding wider phenotypical evaluation of the family history of the patient. Unfortunately, currently familial anamnesis is unavailable for us. We look forward addressing this question in depth in the future. We discuss this issue in the lines 52t-538 of the manuscript. We also feel that existing standard evaluation forms (e.g. Beighton score) would be insufficient to explore subtle collagen-related expressions, so there is need to create special form ad hoc.

Reviewer #2: Authors performed a genetic study on preterm birth due to cervical insufficiency with the support of systematic literature review and NGS analysis of 21 patients based on these review results.

Although the number of patients is too low to make any significant conclusion about cervical insufficiency based on findings, especially for rare variants, I believe this study still be publishable for the purpose of using it as a guide and summary of reviewing the published papers associated with cervical insufficiency. For that purpose, especially for the genetic findings the tone of the paper couldn’t/shouldn’t be too assertive. Authors did already a good job for being NOT assertive for the findings on 21 samples.

Answer: We indeed tried to be as accurate as possible with any statements regarding our NGS findings. We agree that those provide preliminary data and require further evidence in order to be stated in the assertive manner.

In the manuscripts authors indicated that (line number 399-401) “The sequencing resulted in a median coverage depth of 135±38× of the target region, with 94.4% of target regions being covered at least 10 times and 89.7% being covered at least 20 times.”

What about the depth of coverage for rare deleterious variants, if they can mention this in that section, it would be very useful.

Answer: This data is available in the supplementary table S9 column H “Coverage (VAF)” containing detailed information on all rare variants; we put it there due to shortage of space in the Table 3 within the manuscript.

Line 60, "Sention", it would be useful to provide a reference in parenthesis as it is citing the kits. Example:(USA, CA)

Answer: We have now complemented the text with the relevant references.

Line 62, Please cite BWA-MEM algorithm. Although it is very well-known algorithm in the field, for the readers who are not familiar, it is needed to be cited.

Answer: We have now complemented the text with the relevant reference.

Line 75, CADD score threshold was not mentioned, usually it is 15 and higher for pathogenic variants. It would be better to write the threshold, since thresholds for other filters were indicated.

Answer: For prioritization of deleterious variants, we used CADD which is “PHRED-scaled score” with threshold >10 (reported as CADD PHRED in the manuscript line 275) to select top 10% most damaging variants, not the raw CADD score, as it is recommended by the authors (https://cadd.gs.washington.edu/info). In the text we renamed the score to “Phred scaled CADD score” to make it clearer (lines 277-278).

Line 76, PHRED Score >10 is too low, which means probability of 1 in 10 of incorrect base call, usually 30 is and higher is preferred. I would recommend strongly to adjust it to 30 for this.

Answer: Regarding the base calling quality/accuracy, we used variants passing standard filter criteria annotated by the Sentieon exploiting QUAL score, which also takes into consideration PHRED base calling quality metrics. Then we manually filtered those with depth of ≥10 reads with a variant allele frequency of at least 25% further ensuring appropriate variant quality. We have now complemented the text with the relevant information (lines 266-267).

We also would like to have an opportunity to depict the shared senior authorship. Such option is unavailable in the automatic submission system, but we have made changes in the manuscript.

Thank you very much for your time considering our manuscript for publishing in the Plos One Journal. We look forward hearing from you.

With my kind regards,

Ludmila Volozonoka

---

## [Decision Letter · Decision Letter 1]

10 Mar 2020

Genetic landscape of preterm birth due to cervical insufficiency: comprehensive gene analysis and patient next-generation sequencing data interpretation

PONE-D-20-02423R1

Dear Dr. Volozonoka,

We are pleased to inform you that your manuscript has been judged scientifically suitable for publication and will be formally accepted for publication once it complies with all outstanding technical requirements.

With kind regards,

Obul Reddy Bandapalli, MSc, PhD

Academic Editor

PLOS ONE

Additional Editor Comments (optional):

Reviewers' comments:

Reviewer's Responses to Questions

**Comments to the Author**

1. If the authors have adequately addressed your comments raised in a previous round of review and you feel that this manuscript is now acceptable for publication, you may indicate that here to bypass the “Comments to the Author” section, enter your conflict of interest statement in the “Confidential to Editor” section, and submit your "Accept" recommendation.

Reviewer #1: All comments have been addressed

Reviewer #2: All comments have been addressed

2. Is the manuscript technically sound, and do the data support the conclusions?

Reviewer #1: Yes

Reviewer #2: Yes

3. Has the statistical analysis been performed appropriately and rigorously? 

Reviewer #1: I Don't Know

Reviewer #2: Yes

4. Have the authors made all data underlying the findings in their manuscript fully available?

Reviewer #1: Yes

Reviewer #2: Yes

5. Is the manuscript presented in an intelligible fashion and written in standard English?

Reviewer #1: Yes

Reviewer #2: Yes

6. Review Comments to the Author

Reviewer #1: (No Response)

Reviewer #2: (No Response)

7. PLOS authors have the option to publish the peer review history of their article (what does this mean?). If published, this will include your full peer review and any attached files.

Reviewer #1: No

Reviewer #2: No

---

## [Editor Report · Acceptance letter]

12 Mar 2020

PONE-D-20-02423R1 

Genetic landscape of preterm birth due to cervical insufficiency: comprehensive gene analysis and patient next-generation sequencing data interpretation 

Dear Dr. Volozonoka:

I am pleased to inform you that your manuscript has been deemed suitable for publication in PLOS ONE. Congratulations! Your manuscript is now with our production department. 

With kind regards,

on behalf of

Dr. Obul Reddy Bandapalli 

Academic Editor

PLOS ONE